# SOX4 Mediates ATRA-Induced Differentiation in Neuroblastoma Cells

**DOI:** 10.3390/cancers14225642

**Published:** 2022-11-17

**Authors:** Dongyang Zhang, Baocheng Gong, Qiang Zhao, Zhijie Li, Xiaolin Tan, Zhongyan Hua

**Affiliations:** 1Department of Pediatrics, Shengjing Hospital of China Medical University, Shenyang 110004, China; 2Liaoning Key Laboratory of Research and Application of Animal Models for Environmental and Metabolic Diseases, Medical Research Center, Shengjing Hospital of China Medical University, Shenyang 110004, China; 3Department of Pediatric Oncology, Tianjin Medical University Cancer Institute and Hospital, Tianjin 300060, China; 4Department of Rehabilitation, Shengjing Hospital of China Medical University, Shenyang 110004, China

**Keywords:** neuroblastoma, cell differentiation, ATRA, SOX4

## Abstract

**Simple Summary:**

Neuroblastoma (NB) is considered to be caused by the differentiation failure of neural crest cells. Researchers are working on exploring the mechanisms of NB cell differentiation to improve the cure rate. Here, our results show that SOX4 has a significant effect on NB cell proliferation, cells’ neurites, and the cell cycle and that SOX4 mediates the effect of RA in NB cells. All indicate that SOX4 may be a target to induce NB cell differentiation.

**Abstract:**

Neuroblastoma (NB), which is considered to be caused by the differentiation failure of neural crest cells, is the most common extracranial malignant solid tumor in children. The degree of tumor differentiation in patients with NB is closely correlated with the survival rate. To explore the potential targets that mediate NB cell differentiation, we analyzed four microarray datasets from GEO, and the overlapping down- or upregulated DEGs were displayed using Venn diagrams. SOX4 was one of the overlapping upregulated DEGs and was confirmed by RT-qPCR and Western blot in ATRA-treated NGP, SY5Y, and BE2 cells. To clarify whether SOX4 was the target gene regulating NB cell differentiation, the correlation between the expression of SOX4 and the survival of clinical patients was analyzed via the R2 database, SOX4 overexpression plasmids and siRNAs were generated to change the expression of SOX4, RT-qPCR and Western blot were performed to detect SOX4 expression, cell confluence or cell survival was detected by IncuCyte Zoom or CCK8 assay, immunocytochemistry staining was performed to detect cells’ neurites, and a cell cycle analysis was implemented using Flow cytometry after PI staining. The results showed that the survival probabilities were positively correlated with SOX4 expression, in which overexpressing SOX4 inhibited NB cell proliferation, elongated the cells’ neurite, and blocked the cell cycle in G1 phase, and that knockdown of the expression of SOX4 partially reversed the ATRA-induced inhibition of NB cell proliferation, the elongation of the cells’ neurites, and the blocking of the cell cycle in the G1 phase. These indicate that SOX4 may be a target to induce NB cell differentiation.

## 1. Introduction

Neuroblastoma (NB), which derives from neural crest cells, is the most common extracranial malignant solid tumor in children and is considered to be caused by the differentiation failure of neural crest cells [1,2,3]. Patients with low-risk NB have a very good prognosis; however, high-risk patients have consistently poor prognosis. Despite the availability of multiple types of chemotherapy and stem cell transplantation, the long-term survival ratio of high-risk patients is less than 50% [2]. NB is a highly heterogenous tumor, and some tumors can spontaneously heal or differentiation, while others are highly invasive or exhibit a therapeutic resistance phenotype. According to the INSS (International Neuroblastoma Phased System), NB is divided into Phases 1, 2a, 2b, 3, 4, and 4S. Phase 4S NB cells have the potential for differentiation. When the patient is diagnosed, the symptoms are serious, manifesting as multiple tumor metastases; later, the tumor can spontaneously differentiate, and prognosis is good [4]. Another study also found that the degree of tumor differentiation in patients with NB is closely related to its overall survival rate [5]. Therefore, exploring the molecular mechanisms that regulate NB cell differentiation is essential to improve the cure ratio and life quality of patients.

Retinoids are a family of signaling molecules that are related to vitamin A (retinol) in terms of their chemical structures. There are six biologically active isoforms of retinoids: they are all-trans retinoic acid (ATRA); 11-cis retinoic acid; 13-cis retinoic acid; 9, 13-di-cis retinoic acid; 9-cis retinoic acid; and 11, 13-di-cis retinoic acid. Retinoids have been reported to have an effect on cell differentiation, cell proliferation, and cell apoptosis. All-trans retinoic acid (ATRA) is one of the most common retinoids, and it is widely used in the study and treatment of leukemia, lymphoma, neuroblastoma, and glioblastoma [6,7,8]. However, high doses of ATRA are associated with side effects, including teratogenicity and chemical hepatitis [9,10]. Therefore, researchers and clinicians are working on exploring more selective targets and less toxic compounds, which could function as ATRA [7,11,12].

SOX4 is a member of the SOX (SRY-related HMG-box) transcription factor family, which contains a high mobility group (HMG) DNA binding domain (DBD), a glycine-rich domain, and a serine-rich domain [13,14]. Members of the SOX transcription factor family contribute to the development of many organs and tissues, including central nervous system (CNS), retina, bone, hematopoietic system, lymphatic system, and so on. As a member of this family, SOX4 is closely associated with both normal development and various cancers, such as lung cancer, breast cancer, leukemias, glioblastoma, and medulloblastoma [14]. SOX4 also contributes to the differentiation of skeletal myoblast differentiation [15], and divergent sarcomatous differentiation in uterine carcinosarcoma [16]. Studies show that SOX4 cooperates with neurogenin 3 to determine the cell fate during the development of pancreatic beta cells [17], and SOX4 partners with neurogenin 2 to activate Tbrain2 to determine the cell fate of intermediate progenitors during the neuronal differentiation [18]. Studies also show that SOX4 regulates cell survival and metastasis of cancer cells, such as breast cancer [13,19,20].

In this study, we analyzed four microarray datasets from GEO and identified the overlapping differentially expressed genes (DEGs), and the results showed that SOX4 was one of these overlapping upregulated genes, a result that is in accordance with our previous study [21]. Furthermore, our results showed that the survival probabilities of clinical patients have a positive correlation with SOX4 expression, in which overexpressing SOX4 inhibited NB cell proliferation; elongated cells’ neurites; and blocked the cell cycle in the G1 phase and that knockdown of the expression of SOX4 partially reversed the function of ATRA in NB cell proliferation, cells’ neurites, and the cell cycle. All these indicate that SOX4 plays an important role in the differentiation of NB cells, and prompt that SOX4 may be a potential new therapy target for the clinical treatment of patients with NB.

## 2. Materials and Methods

### 2.1. Reagents

All-trans retinoic acid (ATRA) was from Sigma (R2625, Ronkonkoma, NY, USA). Fetal bovine serum was from Gibco (Brisbane, Australia). RPMI-1640 medium, glutamine, antibiotics (penicillin 100 units/mL, streptomycin 100 μg/mL), and PBS were from Bioind (Beit-Haemek, Israel). Trizol was from Solarbio (R1100, China), the GoScript^TM^ Reverse Transcription system was from Promega (A5001, Madison, WI, USA), and TB Green was from TAKARA (RR820A, Beijing, China). Antibody against SOX4 (ab70598) and Goat polyclonal Secondary Antibody to Rabbit IgG-H&L (Alexa Fluor 488, ab150077) were from Abcam, and the antibody against GAPDH and α-Tubulin was from Proteintech (Wuhan, China). Cell counting kit-8 (CCK8) was from Bimake (B34304, Shanghai, China). The cell cycle analysis kit was from Beyotime (C1052, Shanghai, China).

### 2.2. Cell Culture and Treatment

The NB cell lines (NGP, SY5Y, and BE2) used in this study were obtained from Dr. Carol J. Thiele (Cellular and Molecular Biology Section, Pediatric Oncology Branch, National Cancer Institute, National Institutes of Health, Bethesda, MD, USA). All three cell lines (NGP, SY5Y, and BE2) are neuroblastic-type cell lines and can be induced to differentiate. The cell lines were cultured in RPMI-1640 medium with 10% fetal bovine serum, 2 mM glutamine, and antibiotics at 37 °C in 5% CO2 incubator. NGP, SY5Y, and BE2 cells were treated with or without ATRA (5 µM) for 48 h and were collected for RT-qPCR and Western Blotting.

### 2.3. Microarray Data and Data Analysis

Four microarray datasets (GSE45587: NB cell line BE(2)-C cells treated with RA for 24 h and 72 h [22], GSE16451: BE(2)-C cells treated with RA for 3 weeks [23], and GSE87784: Sphere cells from TH-MYCN mice treated with RA for 3 weeks [24]) were from the NCBI Gene Expression Synthesis (GEO) database (http://www.ncbi.nlm.nih.gov/geo/, accessed on 13 January 2021). The volcano plots were generated to display the differentially expressed genes (DEGs) via GEO2R according to the instruction (https://www.ncbi.nlm.nih.gov/geo/info/geo2r.html, accessed on 13 January 2021), and adjusted *p* value (Padj) < 0.05 and |log2(fold change)| > 1 were considered to be statistically significant. The overlapping down- or up-regulated DEGs in the four microarray datasets were displayed using Venn diagrams, which were generated in R language.

The relationship between the expression of SOX4 and clinical patients with NB was analyzed via R2 database (https://hgserver1.amc.nl/cgi-bin/r2/main.cgi?species=hs, accessed on 13 January 2021), and four datasets were used (88, 283, 498, and 649 samples of patients with NB). The relationship between the expression of SOX4 and the differentiation degree of tumors originating from sympathetic nervous system (ganglioneuroblastoma, ganglioneuroma, and neuroblastoma) was analyzed through the Oncomine database (https://www.oncomine.org/resource/login.html, accessed on 13 January 2021).

### 2.4. Cell Transfection

SOX4 expression plasmids were isolated using the HiSpeed Plasmid Maxi Kit (Qiagen, Germany) according to the manufacturer’s instructions. NGP cells were seeded into a 6-well plate (4 × 10^5^/well) and cultured overnight. The SOX4 expression plasmids were transfected into cells using jetPRIME (Polyplus Transfection, Illkirsch, France). After 16 h, cells were collected and seeded into 96-well plates or 6-well plates. After 24 h, the cells were collected for Western Blotting, and after 48 h, they were collected for CCK8 assay, cell confluence analysis using IncuCyte Zoom, immunocytochemistry staining (ICC) assay, and cell cycle analysis.

Small interfering RNAs (siRNAs) (Tongyong, Shanghai, China) were used to knockdown the expression of SOX4. The sequences of siRNAs were SOX4 siRNA #1: 5′-GCAAGCACCUGGCGGAGAATT-3′, 5′-UUCUCCGCCAGGUGCUUGCTT-3′; SOX4 siRNA #2: 5′-GCUGGAAGCUGCUCAAAGATT-3′, 5′-UCUUUGAGCAGCUUCCAGCTT-3′; SOX4 siRNA #3: 5′-CCAACAAUGCCGAGAACACTT-3′, and 5′-GUGUUCUCGGCAUUGUUGGTT-3′. NGP cells were seeded into 6-well plates (4 × 10^5^/well) and cultured overnight. The siRNAs were transfected into cells using jetPRIME. After 16 h, the cells were collected and seeded into 96-well plates or 6-well plates and treated with RA (5 μM), 24 h for RT-qPCR and Western Blotting and 48 h for CCK8 assay, cell confluence analysis using IncuCyte Zoom, ICC assay, and cell cycle analysis.

### 2.5. Quantitative RT-PCR

Total RNA was isolated from NGP cells treated with RA or transfected with siRNAs by using Trizol reagent according to the manufacturer’s instructions and our previous study. The GoScript^TM^ Reverse Transcription System kit was used to generate cDNA. Quantitative PCR was performed with the cDNA by using TB Green according to the manufacturer’s instructions. Beta-actin expression served as an internal control. The relative quantification of gene expression was performed with the 2(−ΔΔCt) method. Details of the PCR primers sequences were as follows: SOX4 sense 5′-GTGGTCCTCAAAGCCAGACACT-3′, SOX4 anti-sense 5′-GCAATGCGCTTTCTGCCGTAGT-3′; β-actin sense 5′- AACT GGGACGACATGGAGAAA -3′, β-actin anti-sense 5′- AGGGATAGCACAGCCTGGATA -3′.

### 2.6. Western Blotting

Total protein was extracted with RIPA Buffer (Beyotime, Shanghai, China) according to the manufacturer’s protocol. The concentration was quantified by using the Bradford reagent (Beyotime, China). Under each condition, 30 μg of total protein was loaded and separated by 10% gel (Epizyme Biomedical Technology Co., Shanghai, China) and then transferred to a PVDF membrane (Millipore, Burlington, MA, USA). The membranes were blocked with 5% skim milk in TBST buffer for 1–2 h and then incubated with primary antibody (SOX4) at 4 °C overnight. The membranes were washed with TBST 3 times and then incubated with the peroxidase-conjugated goat anti-rabbit (1:5000) or anti-mouse (1:5000) antibodies for 2 h at room temperature. The binds were detected using enhanced chemiluminescence (ECL) reagents (Thermo Scientific, Rockford, IL, USA).

### 2.7. Cell Survival Analysis

CCK8 (Cell Counter Kit 8) was used to detect the cell survival, according to the manufacturer’s specification and our previous study. NGP cells were seeded in a 96-well plate 16 h after transfection with SOX4 expression plasmids or siRNAs. CCK8 was added to each well and incubated for 1 h after treated with or without RA 48 h. Optical density was measured at 450 nm using ELISA reader. Cell confluence (%) was calculated and analyzed by using Incucyte Zoom software (Essen BioScince, MI, USA) according to the phase-contrast images, as in our previous study.

### 2.8. Immunocytochemistry Staining

The cell slices were fixed using 4% paraformaldehyde in PBS pH 7.4 for 30 min at room temperature and then washed in PBS 3 times for 3 min. The cell slices were incubated with 0.5% Triton X-100 for 20 min at room temperature and then washed in PBS 3 times. The cell slices were blocked with goat serum for 30 min at room temperature. The goat serum was sucked up with absorbent paper, and the cell slices were incubated with primary antibody (β-Tubulin) overnight at 4 °C. The cell slices were washed in PBST (PBS + 0.1% Tween 20) 3 times for 3 min and incubated with secondary antibody for 1 h at room temperature in the dark. The cell slices were washed in PBST 3 times, and the slices were incubated with DAPI for 5 min at room temperature in the dark. The cell slices were washed in PBST 4 times, the liquid was sucked up with absorbent paper, and the cell slices were observed under fluorescence microscope.

### 2.9. Cell Cycle Analysis

After treated with different conditions, all the cells were harvested, washed with PBS, and then resuspended and fixed with cold 70% ethyl alcohol at 4 °C overnight. Then, the cells were washed again with PBS and incubated with RNase A (100 μg/mL) and PI (50 μg/mL) at room temperature in the dark for 30 minutes. Then, the stained cells were analyzed by Flow cytometry (Becton, Dickinson and Company, Franklin Lakes, NJ, USA). The percentage of cells in each phase of cell cycle was analyzed using the Software of the Flow cytometry system. 

### 2.10. Statistical Analyses

Means ± SD of independent experiments were analyzed by Student’s *t*-test. *p* values less than 0.05 were considered as statistically significant. Data were analyzed by using GraphPad Prism software.

## 3. Results

### 3.1. The Differentially Expressed Genes Obtained from the Microarray Data Analysis

All-trans retinoic acid (ATRA or RA) has been widely used to induce cell differentiation of NB, while the mechanisms still need further study. To explore the potential target gene of RA, we analyzed four microarray datasets from GEO (GSE45587: NB cell line BE(2)-C cells treated with RA for 24 h and 72 h, GSE16451: BE(2)-C cells treated with RA for 3 weeks, and GSE87784: Sphere cells from TH-MYCN mice treated with RA for 3 weeks). The differentially expressed genes (DEGs) were shown in volcano plot form in Figure 1A–C. Additionally, the overlapping DEGs were identified, and there were 13 overlapping upregulated DEGs (Figure 1D) and 2 overlapping downregulated DEGs (Figure 1E). The 13 overlapping upregulated DEGs were SOX4, SOX9, ADD3, ATP7A, CAMK2N1, CTSB, RET, CYP26A1, CYP26B1, DDAH2, LTBP3, MEIS1, and NCOA3 (Table 1). The two overlapping downregulated DEGs were FHL2 and BRCA2 (Table 2).

### 3.2. SOX4 Has a Positive Correlation with the Survival Rate of Patients with NB

SOX4 is a member of the SRY-related HMG-box (SOX) family of transcription factors and has been reported to involved in the regulation of embryonic development and in the determination of cell fate. To identify if SOX4 plays a role in NB, we treated the cells (NGP, SY5Y, and BE2) with RA (5 µM) for 48 h first and then detected the expression of SOX4 at the mRNA level and protein level using RT-qPCR and Western blotting. The results showed that the expression of SOX4 was significantly increased at both the mRNA level and the protein level (Figure 2B,C), which was in consonance with the microarray data from our previous data (Figure 2A) [21] and the other four microarray datasets (Table 1). Then, we analyzed the data with 88, 283, 498, and 649 samples of patients with NB from the R2 database, and the results showed that patients with higher expressions of SOX4 have better overall survival probability and relapse-free/progression-free/event-free survival probability compared to those with lower expressions of SOX4 (Figure 3A–D). To identify if SOX4 was correlated with the differentiation degree of tumors originating from the sympathetic nervous system, we analyzed the expression of SOX4 in ganglioneuroblastoma, ganglioneuroma, and neuroblastoma through the Oncomine database, and the results showed that there have no significant difference between the three tumors (Figure 3E). These results indicated that SOX4 is positively correlated with the survival rate of patients with NB but did not demonstrate the exact relationship between SOX4 and the differentiation degree of sympathetic nervous system-originating tumors.

### 3.3. Overexpression of SOX4 Has the Potential to Induce the Differentiation of NB Cells

RA has been widely used clinically to induce the differentiation of NB tumor cells, and our results showed that RA treatment induced the increased expression of SOX4 and that the expression of SOX4 has a positive correlation with the survival rate of patients with NB. To detect if SOX4 mediated the differentiation of NB cells, we transfected SOX4 overexpression plasmids into NGP cells, and the results showed that the expression of SOX4 was significantly overexpressed (Figure 4A). Additionally, the cell confluence (% of the surface area of cells) or cell survival of SOX4-overexpressed NGP cells was detected by IncuCyte Zoom or CCK8 assay, and the results showed that both the cell confluence and the cell survival were inhibited by the overexpression of SOX4 (Figure 4B–D). The cell survival of SOX4-overexpressed NGP cells was 82.11% compared to the empty-vector transfected cells (100%) (*p* < 0.01) (Figure 4D). These indicated that overexpressing SOX4 decreased the NGP cell survival. One of the important characteristics of differentiated NB cells was elongated neuritic projections. Next, ICC staining was performed to show the morphological changes in NB cells after overexpressing SOX4, and elongated neurites can be observed in SOX4-overexpressed NGP cells (Figure 4E). It has been reported that G1 phase blockage correlates with cell differentiation [25,26], so we detected the cell cycle and compared the percentage of cells in the G1 phase between SOX4-overexpressed and empty-vector transfected NGP cells, and the results showed that the percentage of cells in G1 phase was 63.47% in SOX4-overexpressed NGP cells and 57.91% in empty-vector transfected cells, which indicated that overexpressing SOX4 blocked the cells in the G1 phase (*p* < 0.05) (Figure 4F). Similar effects on cell proliferation, cells’ neurites, and the cell cycle were observed in BE2 cells after overexpressing SOX4 (Appendix A). To confirm the change in the cell cycle in the G1 phase, we performed a Western blot to detect the expression changes in Cyclin D1 and CDK4, both of them are key regulators of the G1 phase. The results showed that the expressions of Cyclin D1 and CDK4 decreased after overexpressing SOX4 (Appendix A) in both the NGP and BE2 cells.

### 3.4. Downregulation of SOX4 Partially Blocks the Function of RA in NB Cells

To further evaluate the role of SOX4 in RA-induced NB cell differentiation, we developed three siRNAs to knockdown the expression of SOX4 and then detected whether downregulated SOX4 could block the function of RA. The results showed that all three siRNAs decreased the expression of SOX4 at the protein level in NGP cells (Figure 5A); furthermore, RA-induced increased expression of SOX4 in NGP cells was downregulated after transfection with SOX4 siRNAs, both at the protein and mRNA levels (Figure 5A,B). Additionally, the cell confluence or cell survival of NGP cells transfected with SOX4 siRNAs or/and treated with RA was detected by IncuCyte Zoom or CCK8 assay, and the results showed that both the decreased cell confluence and the cell survival induced by RA were blocked by SOX4 siRNAs (Figure 5C–E). The cell survival of RA-treated NGP cells was 80.77%, and that of RA-treated and SOX4 siRNA-transfected cells was 100.14% (RA + SOX4 siRNA#1), 105.21% (RA + SOX4 siRNA#2), and 109.82% (RA + SOX4 siRNA#3) compared to the empty-vector transfected cells (100%) (*p* < 0.01) (Figure 5E). Then, ICC staining was performed to show the morphological changes of NGP cells, and the results showed that RA-induced elongated neurites could be blocked by SOX4 siRNAs (Figure 5F). We also detected the cell cycle and compared the percentage of cells in the G1 phase between RA-treated and SOX4 siRNAs-transfected NGP cells, and the results showed that the percentage of cells in the G1 phase was 65.67% in RA-treated NGP cells, 60.30% in RA + SOX4 siRNA#1-treated cells, 61.80% in RA + SOX4 siRNA#2-treated cells, and 61.68% in RA + SOX4 siRNA#3-treated cells (*p* < 0.01) (Figure 5G). These results indicated that knocking down the expression of SOX4 could reverse the function of RA in NB cells. Similar effects on cell proliferation, cells’ neurites, and the cell cycle were observed in RA-treated BE2 cells after the knockdown of SOX4 (Appendix A). To confirm the change in the cell cycle in the G1 phase, we performed a Western blot to detect the expression changes of Cyclin D1 and CDK4. The results showed that the knockdown of SOX4 reversed the decreased expressions of Cyclin D1 and CDK4 induced by RA (Appendix A) in both NGP and BE2 cells.

## 4. Discussion

To explore the potential targets that mediate RA-induced NB cell differentiation, we analyzed four microarray datasets from GEO, which were performed after RA treatment, and then, the overlapping DEGs were identified; there were 13 overlapping upregulated DEGs and 2 overlapping downregulated DEGs. SOX4 was one of the overlapping upregulated DEGs, which is in accordance with our previous study [21], and was confirmed in RA-treated NB cell lines (NGP, SY5Y, and BE2). Our results also showed that the clinical patients’ survival probabilities was positively correlated with SOX4 expression, in which overexpressing SOX4 inhibited NB cell proliferation, elongated the cells’ neurites, and blocked the cell cycle in the G1 phase. Furthermore, knockdown of the expression of SOX4 partially reversed the RA-induced inhibition of NB cell proliferation, the elongation of the cells’ neurites, and the blocking of the cell cycle in the G1 phase. These indicate that SOX4 may be a target to induce NB cell differentiation.

NB is considered to be caused by the differentiation failure of neural crest cells, and it has been reported that the degree of tumor differentiation in patients with NB is closely related to its overall survival rate [1,2]. However, the mechanisms that regulate the differentiation of NB are still not very clear. In this study, we analyzed four microarray datasets from GEO (GSE45587: NB cell line BE(2)-C cells treated with RA for 24 h and 72 h, GSE16451: BE(2)-C cells treated with RA for 3 weeks, and GSE87784: Sphere cells from TH-MYCN mice treated with RA for 3 weeks), and 13 overlapping upregulated DEGs (SOX4, SOX9, ADD3, ATP7A, CAMK2N1, CTSB, RET, CYP26A1, CYP26B1, DDAH2, LTBP3, and MEIS1) and 2 overlapping downregulated DEGs (FHL2 and BRCA2) were identified via Venn diagrams. Both SOX4 and SOX9 are from the SOX family of transcription factors, and SOX transcription factor family members have been reported play important roles in the development of many organs and tissues [13,14]. SOX9 has been reported to be one of the important regulators in neural crest cell development [27], ATP7A plays important roles in neuronal differentiation [28] and glial differentiation [29], CTSB promotes the differentiation of preadipocytes [30], RET has been reported regulate the differentiation of NB cells [31], CYP26A1 and CYP26B1 are retinoic acid catabolic enzymes [32,33], DDAH2 is a biomarker for neural stem cell differentiation [34], LTBP3 regulates the differentiation of mesenchymal stem cells [35,36], and MEIS1 regulates cell proliferation and differentiation during cell fate commitment in different neoplasms [37]. FHL2 has been reported to play important roles in different cells’ differentiation, including neuronal cells, gastric and colon cancer cells, and limb mesodermal progenitors [38,39,40], and BRCA2 regulates the differentiation of both normal tissues and different cancers [41,42,43,44]. These indicate that the overlapping up- and downregulated DEGs from the four GEO datasets may have essential roles in RA-induced cell differentiation.

In this study, we focused on the role of SOX4 in NB cells. As a member of the SOX transcription factor family, SOX4 has been shown to have an essential relationship with not only normal development but also cancers, such as lung cancer, breast cancer, leukemias, glioblastoma, and medulloblastoma [14,15,16]. Firstly, we found that patients with higher expressions of SOX4 exhibited higher overall survival probability and relapse-free/progression-free/event-free survival probability compared to patients with lower expressions of SOX4 by analyzing the data from the R2 database, while the results from the Oncomine database could not demonstrate the exact relationship between SOX4 and the differentiation degree of sympathetic nervous system-originating tumors. Then, our results showed that overexpressing SOX4 inhibited NGP cell proliferation, elongated the cells’ neurites, and blocked the cell cycle in the G1 phase and that the downregulation of SOX4 partially reversed the RA-induced inhibition of NGP cell proliferation, the elongation of the cells’ neurites, and the blocking of the cell cycle in the G1 phase. Guanhua Song et al.’s study evaluated the function of SOX4 in ATRA-induced differentiation of acute promyelocytic leukemia (APL) and, similar to the role of SOX4 in NB cells from our study, showed that SOX4 is essential for the differentiation and regulated by PAD4 [45]. Yuyin Yi et al.’s study found that SOX4 promotes the BMP2regulated differentiation of invasive trophoblast [46], while Wei Han et al.’s study found that FHL3 blocked glioma cell proliferation via the downregulation of SOX4 [47], and Dong Chen et al.’s study shows that the knockdown of SOX4 decreased cell proliferation, migration, and invasion and induced apoptosis in osteosarcoma cell lines [48]. These indicate that SOX4 play different roles in different types of cells. The limitation of this study is that the mechanisms underlying how SOX4 regulates the differentiation of NB cells need to be well-studied, which are the focus of our future studies.

In summary, our present study clearly provided evidence that SOX4 plays an important role in the differentiation of NB cells.

## 5. Conclusions

In conclusion, our study demonstrated that NB patients with higher expressions of SOX4 had good prognosis. Overexpressing SOX4 inhibited NB cell proliferation, elongated the cells’ neurites, and blocked the cell cycle in the G1 phase, and the knockdown of the expression of SOX4 partially reversed the function of RA in NB cells, which provided evidence that SOX4 mediates the differentiation of NB cells.

## Figures and Tables

**Figure 1 cancers-14-05642-f001:**
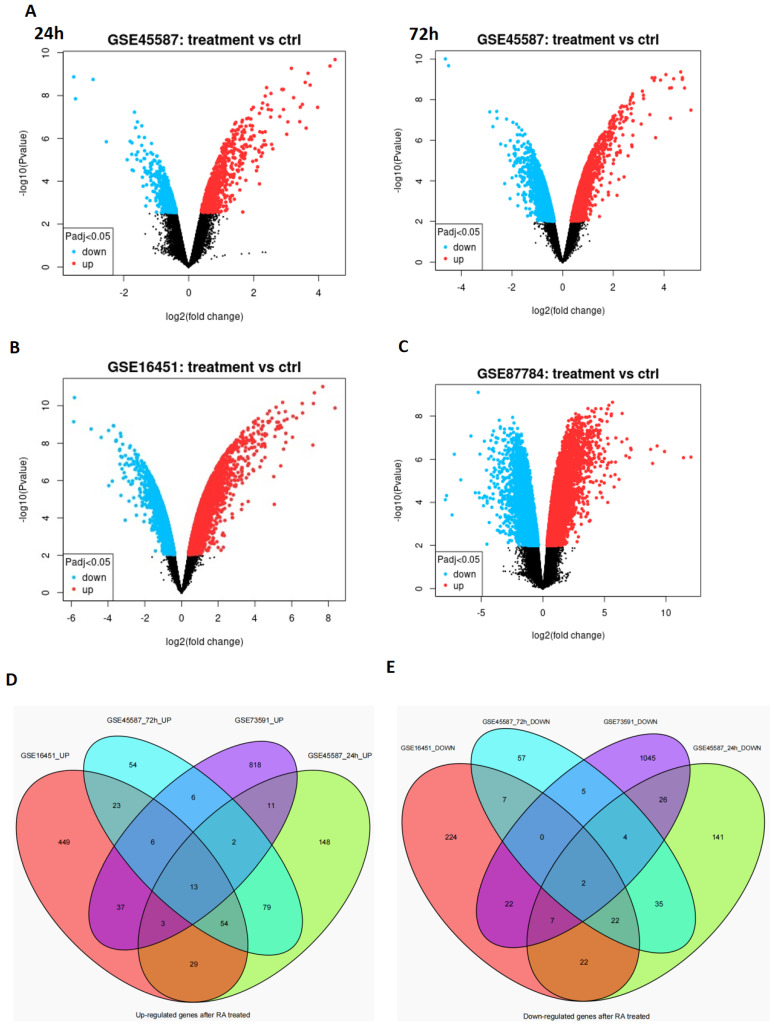
Volcano plot distribution of DEGs and the overlapping DEGs of the four GEO datasets. The volcano plot of (**A**) GSE45587: NB cell line BE(2)-C cells treated with RA for 24 h and 72 h, (**B**) GSE16451: BE(2)-C cells treated with RA for 3 weeks, and (**C**) GSE87784: Sphere cells from TH-MYCN mice treated with RA for 3 weeks. The blue points indicate the downregulated DEGs, red points indicate the upregulated DEGs, and the gray points indicate the genes without significant changes. All DEGs were screened based on an adjusted *p* value < 0.05 and |log2(fold change)| > 1. (**D**) The overlapping upregulated DEGs and (**E**) the overlapping downregulated DEGs.

**Figure 2 cancers-14-05642-f002:**
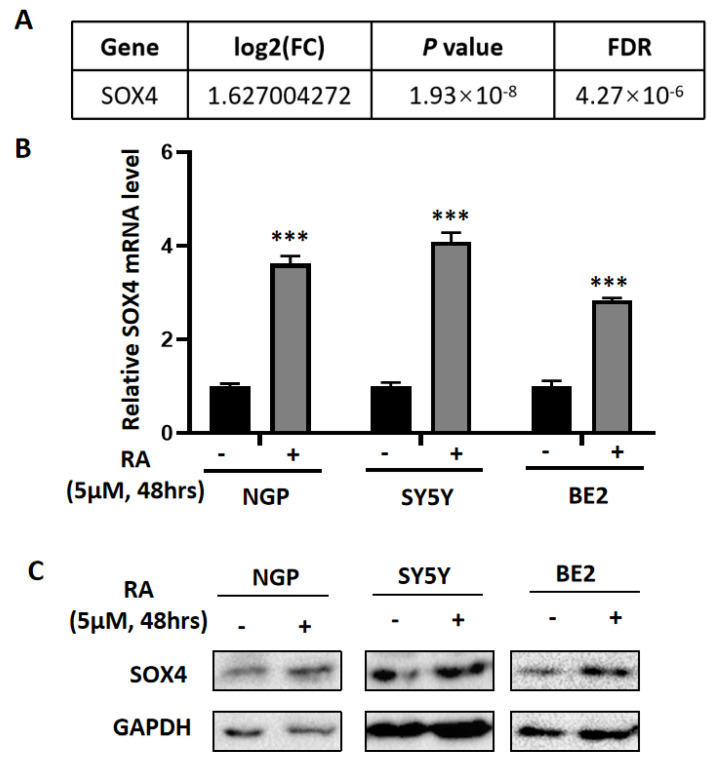
RA treatment increased the expression of SOX4 in NB cells. (**A**) The expression of SOX4 was increased from our previous RNA-Seq data (NGP cells treated with RA for 48 h). RT-qPCR (**B**) and Western blot (**C**) were performed to detect the expression of SOX4 in RA-treated (5 μM, 48 h) NGP, SY5Y, and BE2 cells. Control vs. RA treatment, *** *p* < 0.001. The uncropped western blots have been shown in Appendix A.

**Figure 3 cancers-14-05642-f003:**
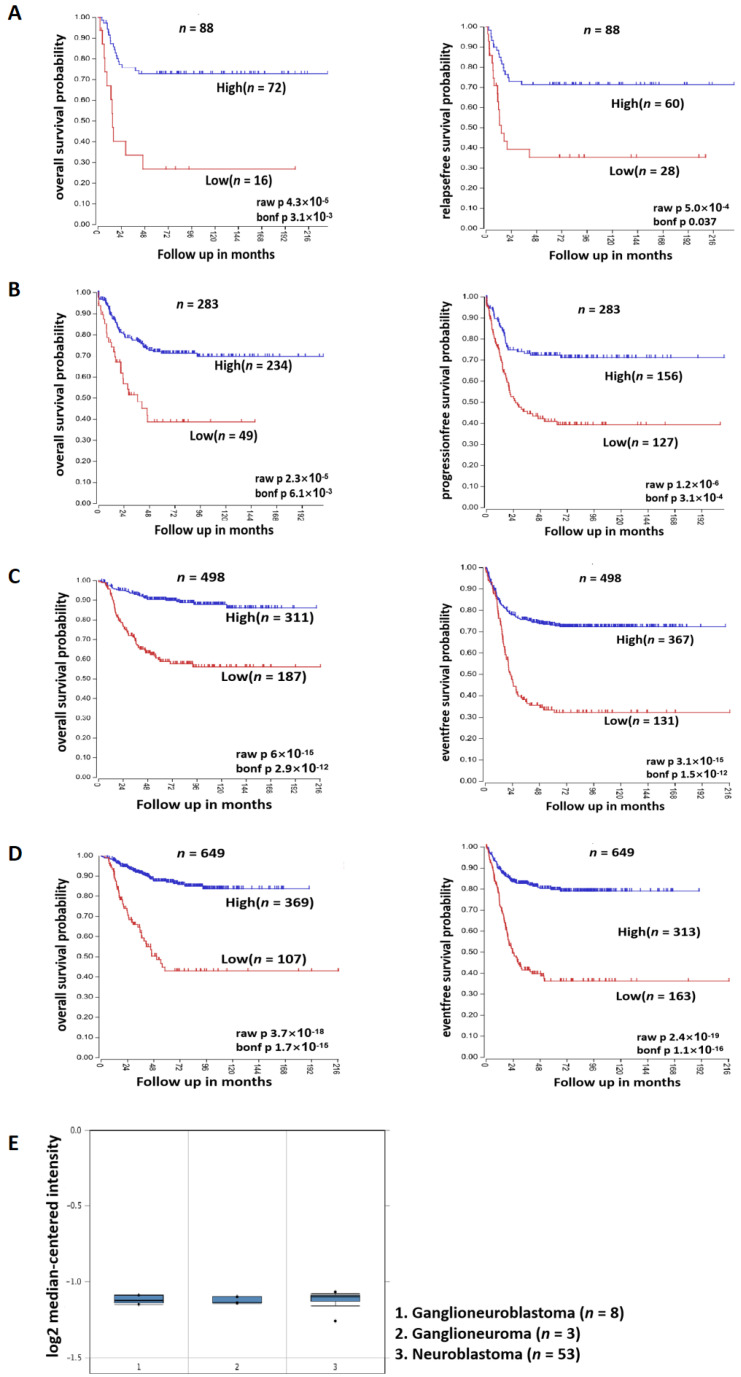
SOX4 has a positive correlation with the survival rate of patients with NB. Four datasets with different clinical samples ((**A**): 88 samples, (**B**): 283 samples, (**C**): 498 samples, and (**D**): 649 samples) are from the R2 database, and the relationship between the expression of SOX4 and overall survival probability and relapse-free/progression-free/event-free survival probability was analyzed. Bonf *p* < 0.05 was considered as statistically significant. (**E**). The relationship between the expression of SOX4 and the differentiation degree of tumors originating from the sympathetic nervous system (ganglioneuroblastoma (*n* = 8), ganglioneuroma (*n* = 3), and neuroblastoma (*n* = 53)) was analyzed through the Oncomine database. Box-plots represent the expression levels of SOX4 (Log2 median-centered intensity).

**Figure 4 cancers-14-05642-f004:**
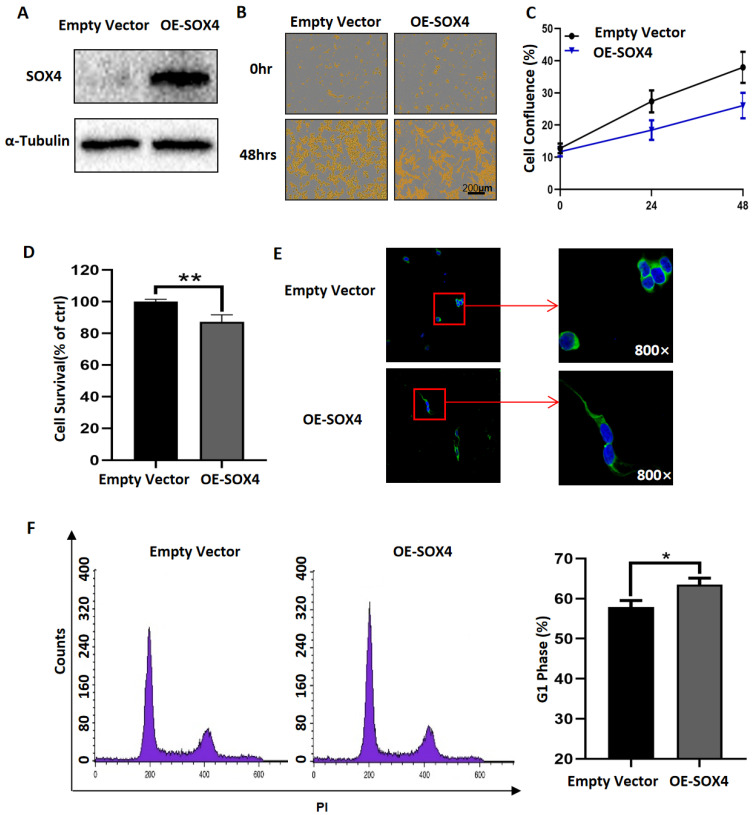
Overexpression of SOX4 has the potential to induce the differentiation of NB cells. (**A**) The expression of SOX4 in NGP cells was detected by Western blot 24 h after transfection with SOX4 overexpression plasmids. The cell confluence (% of the surface area of cells) or cell survival of SOX4-overexpressed NGP cells was detected by IncuCyte Zoom (**B**,**C**) or CCK8 assay (**D**). Empty vector vs. OE-SOX4, ** *p* < 0.01. (**E**) ICC staining was performed to show the morphological changes in NB cells 48 h after overexpressing SOX4. (**F**) Cell cycle analysis was performed 48 h after overexpressing SOX4, and the percentage of cells in G1 phase was analyzed. Empty vector vs. OE-SOX4, * *p* < 0.05. The uncropped Western blots have been shown in Appendix A.

**Figure 5 cancers-14-05642-f005:**
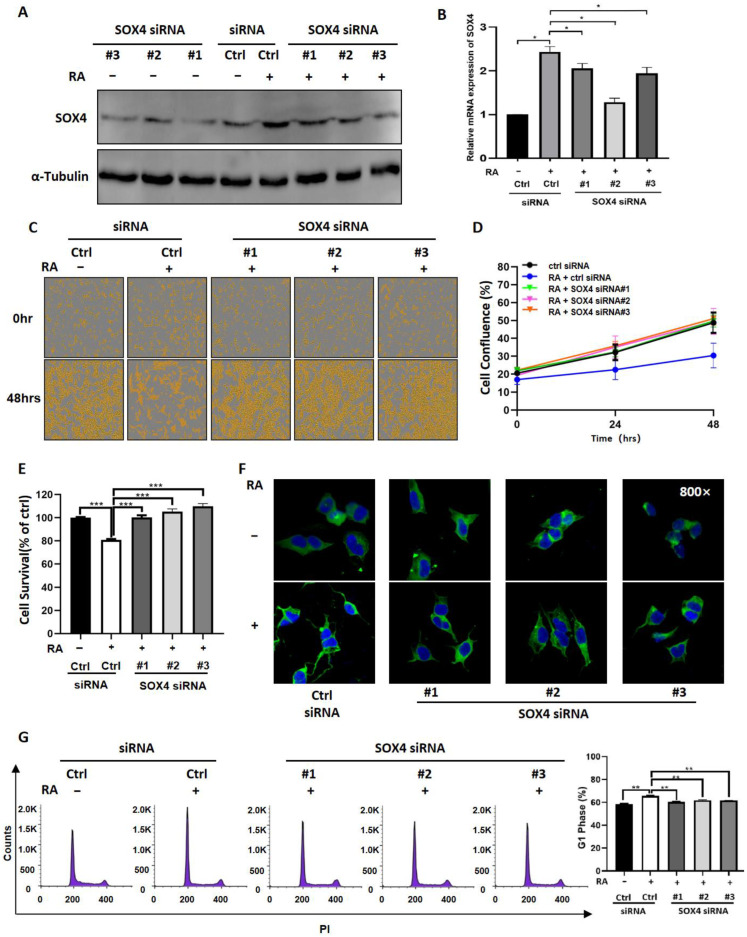
Downregulation of SOX4 partially blocks the function of RA in NB cells. (**A**). The expression of SOX4 in NGP cells transfected with SOX4 siRNAs or/and treated with RA was detected by Western blot (**A**) and RT-qPCR (**B**). Ctrl siRNA vs. Ctrl siRNA + RA, SOX4 siRNA + RA vs. Ctrl siRNA + RA, * *p* < 0.05. The cell confluence or cell survival of NGP cells transfected with SOX4 siRNAs or/and treated with RA was detected by IncuCyte Zoom (**C**,**D**) or CCK8 assay (**E**,**F**) ICC staining was performed to show the morphological changes in NGP cells transfected with SOX4 siRNAs or/and treated with RA. Ctrl siRNA vs. Ctrl siRNA + RA, SOX4 siRNA + RA vs. Ctrl siRNA + RA, *** *p* < 0.001. (**G**) Cell cycle analysis was performed and the percentage of cells in G1 phase was analyzed in RA-treated or/and SOX4 siRNAs-transfected NGP cells. Ctrl siRNA vs. Ctrl siRNA + RA, SOX4 siRNA + RA vs. Ctrl siRNA + RA, ** *p* < 0.01. The uncropped Western blots have been shown in Appendix A.

**Table 1 cancers-14-05642-t001:** Overlapping upregulated DEGs of the four datasets from GEO.

Overlapping Upregulated-DEGs	
SOX4	CYP26A1
SOX9	CYP26B1
ADD3	DDAH2
ATP7A	LTBP3
CAMK2N1	MEIS1
CTSB	NCOA3
RET	

**Table 2 cancers-14-05642-t002:** Overlapping downregulated DEGs of the four datasets from GEO.

Overlapping Downregulated-DEGs
FHL2
BRCA2

## Data Availability

The data presented in this study are available from the corresponding author upon request.

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
