# Peer review of "SOX4 Mediates ATRA-Induced Differentiation in Neuroblastoma Cells"

_cancers, 2022, doi:10.3390/cancers14225642_

Round 1
Reviewer 1 Report
The paper focuses on the role of SOX4 transcription factor in neuroblastoma cell differentiation and growth. The authors have treated two NB cell lines (NGP and SY5Y) with retinoic acid (RA) to promote cell differentiation, and analyzed differential expression genes using GEO database. They identified 13 up-regulated genes (among which SOX4), and 2 down-regulated genes. Increased expression of SOX4 in RA – treated NB cells was confirmed at both RNA and protein levels by RT-qPCR and WB analysis. In NGP-NB cells treated with SOX4 overexpression plasmid, the significant decreased cell survival (82% compared to empty control) has been observed. The reduced survival was accompanied by certain morphological changes, and was ascribed to the blockage of cell division in G1 phase (63% of treated in G1 compared to 58% in empty control). In addition, the authors have shown that the down-regulation of SOX4 reverse the observed effects (reduced cell survival and G1 phase blockage) of RA treatment.
The results confirm the role of SOX4 in the NB cells differentiation, which is in good agreement with previous studies of SOX4 effect on various tumor cells (APL, invasive trophoblastoma, glioma, osteosarcoma).
Questions for authors:
1. In the Introduction you have mentioned that retinoids have low selectivity as differentiation agents, and are hepatotoxic. Could you briefly discuss the alternatives? Why you have selected ATRA?
2. Figure 3, panel E is very difficult to follow and the legend is scarce. What is on the y-axis? How the correlation of SOX4 expression and cell differentiation is represented?
3. Figure 4, panel F – could you briefly comment on the statistical significance of the observed difference between G1-percentage in SOX4 overexpressed cells and empty control? There is 5-6% more G1-phase cells in SOX4-OE – is it enough to confirm G1-phase blockage?
4. You have selected SOX4 among the ATRA up-regulated DEGs. What would happen if you have selected two protein targets from the identified DEGs, and simultaneously transfected plasmids for both? Would it be possible to observe some synergistic effect (for example, in view of reduced cell survival), specific for investigated NB cells?
Reviewer 2 Report
This manuscript is related to SOX4 and neuroblastoma differentiation mechanism and also SOX4 is a promising as a new therapeutic target for neuroblastoma. The authors firstly have been determined the SOX4 expression level in neuroblastoma tissues from bioinformatic data from microarray datasets from GEO. They also analyzed the clinical survival probabilities and SOX4 relationship in the neuroblastoma patients from this data. Then, they have been done some cell proliferation related experiments by overexpression or knockdown of SOX4 in SH-SY5Y neuroblastoma cells by under retinoic acid treatment for differentiation of these cells.
The manuscript has showed that SOX-4 might be a therapeutically target for differentiation of neuroblastoma by induction of expression of SOX4.
Reviewer 3 Report
In my opinion, the article entitled SOX4 mediates the differentiation of Neuroblastoma cells, is a good work and it is suitable for publication in Cancers with minor revision. The results are well showed, documented and the text is properly written. Tables and Figures are clear and well presented. In summary, authors show that SOX4 play an important role in the differentiation of NB cells.
MINOR points:
INTRODUCTION:
I think that Introduction must be improved. The role of SOX 4 in cell fate and differentiation should be better clarified. There are several works related to its role in cancer and metastasis and I think that more information must be added.
MATERIAL AND METHODS:
1. You must describe the cell lines used and the reason why you have selected them.
2. Have you used any antibiotic to select transfected cells?
3. CCK8 is not explained. I have been looking for this term because I did not know it. You should describe CCK8 (Cell Counter Kit 8). Have you also used ELISA reader to detect dead cells?
RESULTS:
1. Have you counted the differentiated cells? I mean have you counted cells with neurites and have you measured their length? I think that these data are important and they would complement the confluence data. You must provide data about differentiated cell number.
2. Why is the reason to choose NGP cells to transfect? Have you performed experiments with SY5Y?
Round 2
Reviewer 1 Report
The authors have properly addressed all my questions, what resulted in the significant improvement of the manuscript.
Author Response
Many thanks again for the comments, which really help us improved the manuscript.